# Prognostic Factors for Cardiotoxicity among Children with Cancer: Definition, Causes, and Diagnosis with Omics Technologies

**DOI:** 10.3390/diagnostics13111864

**Published:** 2023-05-26

**Authors:** Kondylia Antoniadi, Nikolaos Thomaidis, Petros Nihoyannopoulos, Konstantinos Toutouzas, Evangelos Gikas, Charikleia Kelaidi, Sophia Polychronopoulou

**Affiliations:** 1Department of Pediatric Hematology-Oncology (T.A.O.), “Aghia Sophia” Children’s Hospital, Goudi, 11527 Athens, Greece; 2Department of Chemistry, National and Kapodistrian University of Athens, 15772 Athens, Greece; 3First Department of Cardiology, University of Athens, Hippokration Hospital, 11527 Athens, Greece

**Keywords:** cardiotoxicity, childhood cancer, chemotherapeutics agents, biomarkers, omics technology

## Abstract

Improvements in the treatment of childhood cancer have considerably enhanced survival rates over the last decades to over 80% as of today. However, this great achievement has been accompanied by the occurrence of several early and long-term treatment-related complications major of which is cardiotoxicity. This article reviews the contemporary definition of cardiotoxicity, older and newer chemotherapeutic agents that are mainly involved in cardiotoxicity, routine process diagnoses, and methods using omics technology for early and preventive diagnosis. Chemotherapeutic agents and radiation therapies have been implicated as a cause of cardiotoxicity. In response, the area of cardio-oncology has developed into a crucial element of oncologic patient care, committed to the early diagnosis and treatment of adverse cardiac events. However, routine diagnosis and the monitoring of cardiotoxicity rely on electrocardiography and echocardiography. For the early detection of cardiotoxicity, in recent years, major studies have been conducted using biomarkers such as troponin, N-terminal pro b-natriuretic peptide, etc. Despite the refinements in diagnostics, severe limitations still exist due to the increase in the above-mentioned biomarkers only after significant cardiac damage has occurred. Lately, the research has expanded by introducing new technologies and finding new markers using the omics approach. These new markers could be used not only for early detection but also for the early prevention of cardiotoxicity. Omics science, which includes genomics, transcriptomics, proteomics, and metabolomics, offers new opportunities for biomarker discovery in cardiotoxicity and may provide an understanding of the mechanisms of cardiotoxicity beyond traditional technologies.

## 1. Introduction

With the induction of new chemotherapeutic agents over the years, the 5-year survival rate for childhood malignancies exceeds 80%. Caring for these patients includes not only early survival but also later outcomes. Chronic health conditions and health-related quality of life are noted among these, as many long-term treatment-related complications have resulted in increased morbidity and mortality rates. Cardiotoxicity represents the most serious non-hematological toxicity of chemotherapeutic drugs. It is noted that childhood cancer survivors have an 8× higher risk of mortality due to cardiovascular disease and a 6× increased risk of congestive heart failure. Most importantly, early cardiotoxicity may affect the design of chemotherapy and the omission of radiotherapy, resulting in incomplete cancer treatment, and consequently, inferior outcomes. The study of cardiotoxicity among the pediatric population is, as expected, of particular importance due to their long life expectancy [1,2,3,4,5,6].

## 2. Cardiotoxicity

### Definition

The term cardiotoxicity was first described in 1946 as the damage to the heart caused by local anesthetics, mercurial diuretics, and digitalis. Later, in the 1970s, the term broadened to encompass cardiac complications related to anthracyclines (doxorubicin and daunorubicin), combination therapies such as doxorubicin and radiation, and drugs such as 5-fluorouracil. Presently, there is increased research interest, both basic and clinical, in detecting and managing cardiotoxicity as early as possible.

The definition of cardiotoxicity has great significance for a patient’s management. According to the International Cardio-Oncology Society (IC-OS), the cardiovascular complications of chemotherapy can be separated into the following clinical entities and/or categories: (i) cardiac dysfunction: cardiomyopathy/heart failure (HF), (ii) vascular toxicity, (iii) myocarditis, (iv) arterial hypertension, and (v) arrhythmias and QT prolongation [7,8].

The most preponderant diagnosis of cardiotoxicity is based on the changes found in the left ventricular (LV) systolic function measured by the left ventricular ejection fraction (LVEF). Different organizations have defined cardiotoxicity in several ways using different threshold changes in the LVEF [8]. The need to harmonize all these definitions has been met by the International Cardio-Oncology Society (IC-OS) and is supported by the 2022 ESC Guidelines [7,8].

Cardiotoxicity can be categorized according to the time of presentation as acute, early onset, or late onset. Cardiotoxicity can be reversible if addressed while in the early stages [9]. Acute (<1%) toxicity can occur either after administrating a single dose or after a course of chemotherapeutic agents, as long as the onset of clinical manifestations is within the first two weeks following the end of the administration. If presented within the first year of treatment, it is characterized as early onset (1–18%). Late or chronic onset is manifested years or even decades following the treatment [9]. The percentage of late-onset cardiotoxicity varies in the literature mainly due to the different definitions used, the detection methods for cardiotoxicity, the population monitored, and the study design. It seems that over 50% of pediatric cancer survivors showed a subclinical decline in myocardial function and over 16% showed symptoms of clinical HF, especially those who had been exposed to anthracyclines [9].

Abnormalities in ventricular repolarization and electrocardiographic QT-interval alterations, supraventricular and ventricular arrhythmias, acute coronary syndromes, and pericarditis and/or myocarditis-like syndromes are hallmarks of acute or early onset cardiotoxicity [9]. In contrast, asymptomatic systolic and/or diastolic LVD, which can result in dilated cardiomyopathy, is the most typical indicator of chronic cardiotoxicity [9,10]. Clinical and sub-clinical cardiovascular damage, coronary artery disease, and cerebrovascular events are other conditions linked to treatment-related complications. Survivors had an almost six-fold higher risk of heart failure, a five-fold higher risk of myocardial infarction, a six-fold higher risk of pericardial disease, and an almost five-fold higher risk of valvular abnormalities compared to their siblings [11,12,13].

## 3. Chemotherapeutic Drugs

### 3.1. Anthracyclines

Anthracyclines, primarily doxorubicin but also daunomycin, epirubicin, and idarubicin, are some of the most commonly used agents for both hematologic and solid tumors. The basic structure of anthracyclines is that of a tetracyclic molecule with an anthraquinone backbone connected to a sugar moiety by a glycosidic linkage (Figure 1).

Acute cardiotoxicity due to anthracyclines may present as hypotension, tachycardia, arrhythmia, transient depression of left ventricular function, myocarditis, pericarditis, or acute coronary syndrome. Late-onset cardiotoxicity caused by a high cumulative dose of anthracyclines mainly includes signs and symptoms of cardiomyopathy and chronic heart failure [9].

Mitoxantrone is a an anthracenedione (1,4-dihydroxy-9,10-anthraquinon, Figure 1) or anthracycline analog and has similar anthracycline mechanisms of action. Mitoxantrone might cause a wide variety of heart conditions, such as disturbances of cardiac rhythm, chronic heart failure, and persistent diastolic dysfunction in the absence of an impairment of the left ventricular ejection fraction [10].

The prevalent concept of how anthracycline action may cause heart damage involves the production of oxygen radicals, which in turn damage the DNA, proteins, and lipids, leading to cellular dysfunction and myocyte death [14,15,16].

Cardiolipins are abundantly found on the inner mitochondrial cell membrane. By having an increased affinity for anthracyclines, they in turn allow for their increased cell entry. Upon cell entry by passive diffusion, they can reach much higher intracellular concentrations compared to extracellular compartments. Within the cell, they form complexes by binding to iron, thus producing free radicals and reactive oxygen species, which in turn cause cell damage and death. By peroxidizing lipids of the cell membrane, those elements may also damage the cell membrane. As cardiomyocytes contain an abundance of mitochondria, they are more susceptible to anthracycline damage because of the depletion of glutathione peroxidase (an antioxidant) [15].

Other mechanisms of cardiotoxicity include alterations to gene expression and nitric oxide synthase activity, which lead to reduced creatine kinase activity and function in the mitochondria, and ultimately, cell death [15]. After exposure to anthracyclines, many of these subcellular sequelae continue to develop for weeks, shedding light on the mechanisms of chronic cardiomyopathy [14].

Another identified mechanism of doxorubicin-mediated cardiotoxicity is changes to the topoisomerase-II (Top2). Topoisomerase II (TOP2) is a molecule that anthracyclines bind to and inhibit, preventing the growth of tumors. DNA’s phosphate backbone is broken, twisted, and then resealed by topoisomerases, allowing the double helix’s tension to be changed during transcription and replication. Anthracyclines intercalate into DNA, forming complexes with TOP2 that halt the enzyme’s activity and trigger a DNA damage reaction that results in cell death [14,15,17] (Figure 2).

The mechanisms of mitoxantrone-associated cardiotoxicity remain to be completely understood. The formation of reactive oxygen species in myocardial cells is thought to lead to tissue damage through interactions with cellular iron metabolism [10].

### 3.2. Nucleotide Synthesis Inhibitors

The clinical presentation of methotrexate and fluorouracil (5-FU)-induced cardiotoxicity includes myocardial ischemia, cardiogenic shock, heart failure, and cardiomyopathy [10]. Coronary spasm is the most frequently reported mechanism of 5-FU-induced cardiotoxicity (Figure 2) [10]. The data derived from animal models indicate that these chemotherapeutic agents induce oxidative stress and the subsequent apoptosis of cardiomyocytes and endothelial cells [10,17].

### 3.3. Alkylating Agents

Adjuvant DNA-alkylating agents, such as cyclophosphamide (CP) and ifosfamide (IFO), suspend DNA synthesis in cancer cells. These two agents are similar in structure (Figure 3) and engender a similar pattern of cardiotoxic effects, causing acute heart failure, hemorrhagic myopericarditis, and arrhythmia [10,18]. CP- and IFO-induced acute cardiotoxicity is attributed mainly to a rise in free oxygen radicals and a lower antioxidant defense mechanism in the myocardium (Figure 2). A recent study by Sayed-Ahmed MM et al. demonstrated that CP- and IFO-induced cardiotoxicity is due to the inhibition of long-chain fatty acid oxidation via the repression of carnitine palmitoyl transferase I and fatty acid binding protein [19].

### 3.4. Tyrosine Kinase Inhibitors

Dasatinib, imatinib, lapatinib, sorafenib, nilotinib, and sunitinib are examples of small molecule tyrosine kinase inhibitors (TKIs) that suppress cancer cell proliferation and induce apoptosis of cancer cells. Imatinib, dasatinib, and nilotinib are the three FDA-approved TKIs for use as first-line chronic myeloid leukemia therapy in pediatrics. In addition, sorafenib is used in young adults. The pathophysiological mechanism of ΤΚΙ-induced cardiotoxicity is mitochondrial impairment and cardiomyocyte apoptosis (Figure 2) [10,18,20]. Each of the above drugs is associated with a different type of cardiotoxicity. For example, dasatinib is more often associated with pleural effusion and less with hypertension, HF, pericardial effusion, and pulmonary hypertension. Nilotinib is associated with peripheral artery disease, hypertension, and prolonged QTc. In contrast, imatinib is related to less cardiotoxicity than the other TKIs [7].

### 3.5. Anti-Microtubule Agents

Anti-microtubule agents, including docetaxel, paclitaxel, and vinca alkaloids, prevent the polymerization or depolymerization of microtubules [17]. The clinical features of cardiotoxicity induced by anti-microtubule agents are mostly ischemia and arrhythmia. Among all anti-microtubule agents in clinical use, paclitaxel induces the release of histamine, which in turn activates specific cardiac receptors, raising the myocardium’s oxygen need, and leading to coronary vasoconstriction (Figure 2). In addition, Zhang et al. reported that the frequency of spontaneous calcium concentration in cardiomyocytes was significantly increased after paclitaxel treatment. This finding could be of great significance, as fluctuations in blood calcium levels are linked to arrhythmogenesis [10,21].

### 3.6. Cisplatin

Cisplatin is an efficacious chemotherapeutic drug with a strong antitumor effect against a wide range of neoplasms (Figure 4). However, the drug’s acute and cumulative cardiotoxicity, including electrocardiograph (ECG) abnormalities, angina and acute myocardial infarction, hypertension and hypotension, arrhythmias, myocarditis, cardiomyopathy, and congestive heart failure, is a significant factor that restricts cisplatin treatment. Cisplatin cardiotoxicity can be caused by reactive oxygen species generation, which leads to the creation of oxidative stress and endothelial capillary damage (vascular damage) or has a direct toxic effect on cardiac myocytes (Figure 2) [10]. The prolonged cardiovascular toxicity of cisplatin, lasting up to many years, has been explained by both direct diffuse endothelial damage and an increase in risk factors for cardiovascular disease. These effects include coronary artery disease, systolic or diastolic left ventricular dysfunction, and severe congestive cardiomyopathy [10].

### 3.7. Monoclonal Antibodies

Monoclonal antibodies, including bevacizumab (Avastin) and trastuzumab, not used in children, inhibit angiogenesis. Bevacizumab blocks vascular endothelial growth factor (VEGF), while trastuzumab inhibits human epidermal growth factor receptor 2 (HER2) in cancer cells (Figure 5). Bevacizumab causes mostly hypertension, congestive heart failure, and thromboembolic events of the artery and vein through the mechanism of oxidate stress induced by cardiomyocyte apoptosis. Monoclonal antibodies are not widely used in children with malignancy, so we do not discuss them further [10,18,20].

### 3.8. Proteasome Inhibitors

A new therapeutic option for the treatment of acute lymphoblastic leukemia (ALL) includes proteasome inhibitors (Figure 6) [18,20]. Bortezomib and carfilzomib are two newly prescribed drugs with the potential to cause cardiac dysfunction [22]. Compared to carfilzomib (up to 25%), bortezomib has a lower incidence of heart failure (up to 4%). The pathogenesis of proteasome inhibitor cardiotoxicity is not currently well understood. Exposure to proteasome inhibitors in a prenatal mouse model has shown that they can induce oxidative stress, leading to myocardial dysfunction. Carfilzomib is also known to induce renal toxicity and microangiopathy as a consequence of endothelial dysfunction. Combining these studies reveals a complicated mechanism of cardiotoxicity linked to proteasome inhibitors, including alterations to the heart’s muscle and vasculature, which may be more severe with carfilzomib than bortezomib due to the irreversible nature of the proteasome inhibition of carfilzomib [23,24].

## 4. Risk Factors

A multivariable number of factors contribute to the onset of cardiotoxicity with regard to the patient and/or the administered therapy (Figure 7). Several types of chemotherapeutic drugs may cause cardiotoxicity, as referred to above. These drugs act on cancer cells through a variety of mechanisms and promote cardiotoxicity with distinctive clinical symptoms and underlying mechanisms (Table 1) [7,10,16].

The overall dose and mode of administration of each chemotherapeutic agent play an aggravating role in causing cardiac damage. For example, anthracycline-induced cardiotoxicity is known to be both cumulative and dose-related, indicating that each administered dose induces sequential or additional damage [18,26]. The cumulative total anthracycline dose is the most important risk factor for cardiac dysfunction [27]. In retrospective research, Von Hoff et al. [28] observed that when a patient receives a combined doxorubicin dose of 400, 550, and 700 mg/m^2^, the incidence of cardiotoxicity is 3, 7, and 18%, respectively, with dose-limiting toxicity. Another study in adolescents found that even at dosages of 180–240 mg/m^2^, 30% of the participants experienced subclinical episodes 13 years after therapy [29]. These results imply that there is no anthracycline dose that is considered safe. Reduced cardiac function has been correlated with dosages as low as 100 mg/m^2^ [30,31,32].

In addition, female gender, age (under <5 years old), the patient’s clinical condition (extent of disease, infection), genetic background, pre-existing cardiac disease, and the combinations of cardiotoxic drugs play an important role in causing cardiac damage [33] (Table 2). The pediatric population is more homogeneous as a study population since there are no confounding cardiovascular risk factors (diabetes, smoking, arterial hypertension) [7,14,34].

The risk of developing cardiotoxicity is also increased by concurrent radiation exposure to the chest. In addition to the myocardium, radiation therapy has the potential for damaging the pericardium, heart vessels, and conductive tissue [18].

The carriers of certain genetic mutations are also more susceptible to cardiotoxicity [9,18]. Our understanding of genetic susceptibility to anthracycline-related cardiotoxicity has been influenced by a sizable body of research, as we describe below.

## 5. Diagnosis

As part of the baseline risk assessment, a thorough clinical history and physical examination are advised. Many patients with cardiac dysfunction could be asymptomatic, so both during and post-chemotherapy, cardiac monitoring is necessary. Consideration of the classic cardiovascular disease risk factors already mentioned is recommended, and children should be monitored for clinical signs and potential indicators of cardiotoxicity. The above-mentioned factors should be noted alongside the baseline electrocardiography, cardiac serum biomarkers, and cardiac imaging tests to complete baseline evaluation.

### Imaging

Diagnostic approaches for chemotherapy-induced cardiotoxicity include electrocardiography and echocardiography, which are used as methods of monitoring cardiac function before, during, and after treatment.

Electrocardiography (ECG) can be used to identify any early signs of cardiac toxicity, such as resting tachycardia, ST-T wave abnormalities, conduction disturbances, QT interval prolongation, or arrhythmias. However, these ECG findings could be induced by several factors unrelated to cardiotoxic treatment. These ECG abnormalities may be reversed and are not always related to the development of chronic cardiomyopathy [7,27,34].

Two-dimensional echocardiography (2D) is the most used imaging technique to monitor cardiac function. It is non-invasive, cheap, readily available, and does not expose the patient to further radiation. However, standard echocardiographic parameters such as LVEF may lack sensitivity for the detection of systolic dysfunction [34].

Considering the poor sensitivity of 2D LVEF measurement, the use of global systolic longitudinal myocardial strain (GLS) analysis has become an area of interest [34,35]. A pathogenic percentage reduction of GLS greater than 15% from baseline is regarded as a sign of early LV dysfunction. When possible, it is preferable to use these sophisticated echocardiographic measures as the foundation for clinical decisions.

Other methods for the monitoring of these patients are cardiac magnetic resonance imaging (CMR), nuclear cardiac imaging (MUGA), and myocardial perfusion imaging (MPI). There are two techniques for MPI: single-photon emission computed tomography (SPECT) and positron emission tomography (PET). All these methods take considerably longer than follow-up echocardiography and might not be as accessible in all pediatric facilities [34].

## 6. Biomarkers

Several biomarkers have been assessed for their efficacy in the early prediction of patients’ risk of cardiotoxicity and the identification of cardiac dysfunction. The World Health Organization defines biomarkers as any element, structure, or process that can be detected in the body (or its byproducts) and which affects or forecasts the development or course of a disease.

According to the literature, troponin and natriuretic peptide are the most studied biomarkers for the detection of both early cardiotoxicity and its later follow-up. Lipshultz et al. showed that the elevation of cardiac troponin T and N-terminal pro-brain natriuretic peptide (NT-pro-BNP) in children with acute lymphoblastic leukemia was associated with a notably reduced left ventricular (LV) mass, abnormal LV end-diastolic posterior wall thickness, and abnormal LV thickness-to-dimension ratios, all of which suggested LV remodeling, respectively, 4 years later [36]. However, further research has not pointed out an association between acute or chronic troponin release and left ventricular dysfunction, but in contrast, an association has been found with NT-pro-BNP in childhood cancer survivors [37,38,39].

We should be especially careful in evaluating troponin and natriuretic peptide values in children <1 year of age due to their normally elevated values at these ages [40].

Other biomarkers investigated include inflammation markers, such as C-reactive protein (CRP) and growth/differentiation factor 15 (GDF-15) [38], oxidative stress markers such as myeloperoxidase, vascular remodeling markers such as placental growth factor and soluble Fms-like tyrosine kinase receptor 3, and fibrosis markers (galectin 3) [36,41,42,43,44,45,46,47,48,49]. Moreover, these conventional biomarkers usually show significant changes only after heart damage occurs.

To determine the proper use of these biomarkers in clinical practice, new prospective and multicenter studies with large populations, well-standardized dosing methodologies, well-defined time of sampling, and cardiologic end points are required.

## 7. Omics

In the last decades, new research and clinical studies have attempted to identify possible biomarkers of early cardiac damage by chemotherapeutic agents using omics technology. Omics science offers new opportunities for biomarker discovery in cardiotoxicity and may provide an understanding of cardiotoxicity beyond traditional technologies. Omics technology includes genomics, transcriptomics, proteomics, and metabolomics.

### 7.1. Genomics

A cumulative anthracycline dose and other related risk factors seem to not be exclusive risk factors responsible for significant individual variation in the incidence and severity of heart failure in pediatric cancer survivors. Several studies have revealed how important host genetic polymorphisms could lead to a differential risk of cardiotoxicity among cancer survivors with otherwise identical clinical and treatment-related risk factors by using genome-wide association or candidate gene approaches [50,51,52,53]. This explains why some patients experience cardiotoxicity while other patients can tolerate high doses of chemotherapy without heart damage.

Genomic polymorphisms are small changes in a specific part of the DNA chain. One or more polymorphisms can determine a range of patient characteristics, such as their ability to metabolize and eliminate genotoxic substances. Cancer treatment-related cardiovascular toxicity risk may be influenced by genetic variation. Significant efforts using targeted and whole genome correlation studies have been made to reveal the pharmacogenomic causes of this predisposition [50,54,55,56,57,58,59,60].

At least 45 SNPs located in 34 genes have been associated with anthracycline-induced cardiotoxicity [61,62,63,64]. Many of these associations require further investigation through replication and/or functional and mechanistic studies to make sure we confirm and better understand the roles of these associated variants in anthracycline-related cardiotoxicity (ACT) [9].

Polymorphisms in solute carrier transporter (SLC) genes are associated with ACT. One of the functions of the SLC family is acting as drug transporters for anthracyclines, thus providing biological support for these genetic associations. Research on childhood cancer survivors has discovered correlations between ACT risk and protective variants in SLC, such as SLC28A3, SLC22A17, and SCL22A7. These findings have been successfully replicated [65,66,67,68,69,70]. In addition, different studies have reported protective variants in SLC10A2 and SLC22A1 [66]. SLC22A6 was first mentioned in the context of ACT by Sagi et al. in patients treated for childhood ALL [68].

Retinoic acid receptor gamma (RARG) has been involved in cardiac development and remodeling through the repression of Top2b [71]. A recent genome-wide association study, by Aminkeng et al. [51], uncovered a non-synonymous variant rs2229774 in *RARG*, which was significantly associated with ACT in survivors of childhood cancer. Specifically, rs2229774-carriers had a significantly increased risk of developing ACT as compared to non-carriers [51].

Studies have also revealed an elevated risk brought on by a variation in the UGT1A6 gene, a member of the glucuronosyl transferase family. Through the glucuronidation path, UGT1A6 plays a significant role in the detoxification of drugs, including the metabolites of anthracyclines [51,65,69].

Polymorphisms in adenosine triphosphate-binding cassette transporter (ABC) genes are related to cardiotoxicity in childhood patient cancers treated with anthracyclines. The ABC genes seem to play a role as efflux transporters of drugs, including anthracyclines, so may have important effects on the myocardium. Eight variants in five genes (ABCB1, ABCB4, ABCC1, ABCC2, and ABCC5) have been associated with cardiotoxicity, especially with reduced ejection fraction [54,55,56,68,72].

Other studies have investigated polymorphisms in carbonyl reductase genes, which have been associated with dose-dependent increases in cardiomyopathy risk. Carbonyl reductase (CBR) will reduce anthracyclines to cardiotoxic alcohol metabolites. As Blanco et al. showed, among childhood cancer survivors, homozygosity for the G allele in CBR3 leads to increased cardiomyopathy risk associated with low- to moderate-dose anthracyclines. Patients homozygous with the CBR3 V244M G allele have no safe cut-off minimum dose [57,58].

A recent study showed a gene–environment interaction between a single-nucleotide polymorphism on the CELF4 gene and a higher dose of anthracyclines [59]. CUGBP Elav-like family member 4 CELF4) protein is responsible for pre-mRNA alternative splicing of TNNT2, the gene that encodes for cardiac troponin T.

Aminkeng et al. [69] gathered evidence-based clinical practice recommendations for pharmacogenomic testing and emphasized that the RARG genes rs2229774, UGT1A6 * 4 rs1786378, and SLC28A3 rs7853758 have the potential to further discriminate patients at higher and lower risk of ACT. A pharmacogenetic test for these genetic variations in RARG, SLC28A3, and UGT1A6 has been released at the British Columbia Children’s Hospital since the publication of these guidelines. Based on genetic and clinical risk variables, tested patients were divided into several risk groups, and therapy adjustments were chosen in accordance with this risk. Early evidence indicates that the British Columbia Children’s Hospital’s pharmacogenetic testing was effective in lowering the incidence of ACT in children, which should inspire additional clinics to utilize this pharmacogenetic test.

These findings might help develop prediction models that can spot patients who will be particularly susceptible to ACT and who will need their therapy modified or closer monitoring. Further independent research may make it possible to identify people before treatment with a genetic predisposition to cardiovascular toxicity and for whom more thorough screening, or perhaps preventive measures, should be implemented. Replication analyses, however, have occasionally failed to support the initial findings. Numerous factors, including the variability of cohorts, ambiguities in the definition of ACT, variations in the procedures, and the type or dosage of the chemotherapeutic drugs used may have contributed to this. To increase the diagnostic and prognostic role in predicting ACT, more research is required.

### 7.2. Transcriptomics

Another interesting area is the integration of microRNAs in the early detection of cardiotoxicity. Recently, the potential use of circulating microRNAs (miRNAs) has been studied as a possible specific biomarker and therapeutic target of cardiac disease [73,74,75,76,77,78,79].

MicroRNAs are small endogenous non-coding RNAs of 21–24 nucleotides, acting as post-transcriptional gene regulators by inhibiting and/or degrading target messenger RNAs (mRNAs). Bioinformatics data suggest that each miRNA molecule can control hundreds of gene targets, thus indicating the potential effect of miRNAs on virtually any genetic pathway. MiRNAs play a significant role in different biological processes, including proliferation, differentiation, development, and cell death. Furthermore, several miRNAs are involved in regulating heart development from the embryonic to the adult stage, and their dysregulation leads to various heart diseases, such as arrhythmias, essential hypertension, heart failure, cardiomyopathy, cardiac hypertrophy, and atherosclerosis [80,81].

The cardiotoxic effect of chemotherapeutic agents may lead to specific miRNAs with changed expressions. These can be used to investigate the toxicity of potential drug candidates on cardiomyocytes and cell lines originating from the heart in a preclinical in vitro setting. The potential use of circulating miRNAs in plasma as indicators of drug-induced cardiotoxicity has undergone much research during the last several years [80].

Nearly 30 circulating miRNAs have had their levels altered, both increased and decreased, and these changes have been linked to HF and associated pathologies. MiRNAs, including miR-1, miR-133, miR-208a/b, miR-499, miR-29, and miR-34, which are substantially expressed in the myocardium compared to other tissues, are the ones that are primarily being researched [73]. In addition, a variety of harmful substances alter the miRNA profile in both plasma and cardiac tissue. Even at low toxin concentrations, where other tissue damage biomarkers are not discernible, alterations in miRNAs can be measured [80]. Most studies use data from experimental animals, while those utilizing clinical patient samples are limited.

MiR-1 is a skeletal muscle-specific miRNA that has an important role in cardiac development, function, and disease. Abnormal miR-1 levels are associated with acute myocardial infarction, heart failure, arrhythmias, ventricular dysfunction, cardiac hypertrophy, and myocyte hyperplasia [82]. MiR-499 and miR-208 are associated with acute myocardial infarction and HF [82]. Circulating levels of miR-133a have been associated with an increased risk of cardiovascular diseases. Increased levels of miR-133a have been detected in patients with acute myocardial infarction earlier than cardiac troponin T increase [83]. MiR-133 includes two miRNAs, named miR-133a and miR-133b, that are highly expressed in the human heart and seem to be involved in heart development and myocyte differentiation.

The analysis of circulating miRNAs in breast cancer patients receiving doxorubicin (DOX) identified miR-1 as a potential candidate for the early detection of DOX-induced cardiotoxicity [84]. Leger et al. investigated other possible markers of cardiotoxicity in children and young adults treated with anthracycline chemotherapy (AC). Candidate plasma profiling of 24 miRNAs was performed in 33 children before and after a cycle of AC or non-cardiotoxicity chemotherapy. ΜiR-1, miR-29b, and miR-499 were reported to be upregulated in pediatric patients following the acute initiation of AC [85,86]. Monitoring the plasma levels of miR-208a and miR-208b showed an elevation in patients with myocardial damage and were even detected earlier than cardiac troponins [87]. This is in concordance with the findings of other studies [73,74,87,88]. Table 3 provides a summary of the major miRNAs linked to drug-induced cardiotoxicity.

In addition to anthracyclines, other cytotoxic agents have shown cardiotoxic effects, and biomarkers of their pathomechanism have been searched for, including miRNAs. Patients with bevacizumab-induced cardiotoxicity, when compared to controls, were found to have increased levels of five miRNAs. In the validation experiments, two of these (miR-1254 and miR-579) showed valuable specificity. MiR-1254 exhibited the strongest correlation with the clinical diagnosis of bevacizumab-induced cardiotoxicity [89].

With regard to a number of features of drug-induced cardiotoxicity, miRNAs appear to be a promising agent. A potentially successful method for preventing severe problems is the identification of patients with subclinical cardiotoxicity through the detection of cardio-specific miRNAs circulating in plasma that are not present under normal circumstances [80]. Many other research studies should focus on how the miRNA profile changes when interacting with drugs with proven cardiotoxicity.

### 7.3. Proteomics

The proteomic data available to date on chemotherapy-induced cardiac toxicity are limited, mainly involving anthracyclines, and related to experimental animal studies [90].

Proteomics is the study of proteins, which are essential components of organisms and have a variety of functions. The proteome consists of all the proteins expressed by a cell, tissue, or organism. Proteomics could give us important information for a number of biological problems.

Ohyama et al. identified cellular processes in mouse heart tissue from control rats and rats affected by different adriamycin and docetaxel dosing protocols using a toxicoproteomic approach. They identified nine different proteins that were expressed in the control and two treatment groups, and which were involved in energy production pathways, such as glycolysis, the Krebs cycle, and the mitochondrial electron transport chain [91].

Kumar et al. in 2011 used a rat model of doxorubicin-induced cardiotoxicity to show the differential regulation of several key proteins, including proteins that are stress-responsive (ATP synthase, enolase alpha, alpha B-crystallin, translocation protein 1, and stress-induced phosphoprotein 1), and apoptotic/cell damage markers (p38 alpha, lipocortin, voltage-dependent anion-selective channel protein 2, creatine kinase, and MTUS1) [86].

More recently, Desai et al. pinpointed possible biomarkers of early cardiotoxicity in the plasma of male B6C3F1 mice that received a weekly intravenous dose of 3 mg/kg doxorubicin (DOX) or saline (SAL) for 2, 3, 4, 6, or 8 weeks (corresponding to cumulative doses of 6, 9, 12, 18, or 24 mg/kg DOX). They suggested the neurogenic locus notch homolog protein 1 (NOTCH1) and von Willebrand factor (vWF) as early biomarkers of DOX cardiotoxicity to address the clinically significant question of identifying cancer patients at risk for cardiotoxicity [92].

Finally, Yarana et al., using a mouse model of DOX-induced cardiac injury, quantified serum extracellular vehicles (EVs), assayed proteomes, counted the oxidized protein levels in serum EVs generated following DOX treatment, and examined the alteration of EV content. The release of EVs containing brain/heart glycogen phosphorylase (PYGB) before the increase in cardiac troponin in the blood following DOX therapy suggests that PYGB is an early indicator of cardiac damage, according to the proteomic profiling of DOX_EVs [93].

To find out if these pathways could result in the discovery of early markers of cardiotoxicity, more research in this area is required.

### 7.4. Metabolomics

Metabolomics is an upcoming new science with the potential to further increase our knowledge of cancer biology and the search for prognostic biomarkers. Up to now, most studies have either used metabolomic data from experimental animals or the cellular level, while those utilizing clinical patient samples have been extremely limited (Figure 8).

Metabolism is more directly related to the phenotype and physiology of a biological system. Metabolomics is the study of all cellular metabolites (hydrocarbons, amino acids, sugars, fatty acids, organic acids, steroids, and peptides). It encompasses all levels of cellular regulation, that is, the regulation that occurs at the level of transcription, translation, and post-translational modifications, and hence, it can closely reflect the phenotype of an organism at a specific time. The human metabolome is thought to be composed of about 3000 endogenous metabolites at current estimates (the Human Metabolome Project). However, the exact size of the human metabolome is still debatable. It is also believed that nutritional compounds, xenobiotics, and microbial metabolites must be considered when defining the human metabolome [94]. Therefore, metabolome analysis can be a useful tool used to find diagnostic markers that will help us examine unknown pathological conditions effectively.

Different analytical techniques can be used in the measurement of metabolites. Such methods are nuclear magnetic resonance (NMR) spectrometry, molecular mass spectrometry (MS), gas chromatography (GC), high-performance liquid chromatography (LC), and tricarboxylic acid (TCA). The most common and higher-throughput technologies are nuclear magnetic resonance (NMR) spectrometry and molecular mass spectrometry (MS).

Mass spectrometry is an analytical platform for metabolomic analysis. It is a highly sensitive, reproductive, and versatile method, as it identifies molecules and their fragments by measuring their masses. This information is obtained by measuring the mass-to-charge ratio (*m*/*z*) of ions that are produced by inducing the loss or gain of a charge from a neutral species. The sample, which comprises a complicated mixture of metabolites, can be introduced to the mass spectrometer either directly or preceded by a separation approach (using liquid chromatography or gas chromatography) [95].

NMR spectroscopy utilizes the magnetic properties of nuclei to determine the number and type of chemical entities in a molecule. Proton NMR spectroscopy can detect soluble proton-containing molecules with a molecular weight of approximately 20 kD or less. The NMR spectra serve as the raw material for pattern recognition analyses, which simplifies the complex multivariate data into two or three dimensions that can be readily understood and evaluated. Both NMR and liquid chromatography–mass spectrometry (LC-MS) systems can be integrated into in vivo tissues or biological fluids, such as serum, plasma, urine, etc., obtained from humans. The advantages of NMR are that it requires relatively little sample preparation, it is non-destructive, and it can provide useful information regarding the exact structure of metabolites. However, NMR sensitivity is related to the magnet’s strength, while available instrumentation can unambiguously detect only the most abundant metabolites in plasma. On the other hand, the most important advantage of mass spectrometry coupled with upfront chromatography is its far greater sensitivity than NMR MS-based systems, which have been used to resolve compounds in the nanomole to the picomole and even the femtomole range, whereas the identification of compounds by 1H-NMR requires concentrations of 1 nanomole or higher [96,97].

The main methodologies that are used for metabolomic analysis are untargeted and targeted metabolomics. Untargeted metabolomics allow for measuring a wider variety of metabolites present in an extracted sample without prior knowledge of the metabolome. The main advantage is that this provides an unbiased way to examine the relationships among interconnected metabolites from multiple pathways. In contrast, targeted metabolomic analyses measure the concentrations of a predefined set of metabolites and provide higher sensitivity and selectivity than untargeted metabolomics.

An overview of the main metabolomics associated with drug-induced cardiotoxicity detected in plasma/stem cells/hearts in mice and people is given in Table 4. The role of carnitine in the detection of cardiotoxicity was confirmed by a successive study in which Armenian et al. compared a metabolomics analysis in 150 symptom-free childhood cancer survivors who received anthracycline treatment. Thirty-five participants were found to have cardiac dysfunction without symptoms. So, they compared two groups (participants with cardiac dysfunction and those with normal systolic function) and discovered 15 metabolites differentially expressed among the patients. After adjusting for multiple comparisons, individuals with cardiotoxicity had significantly lower plasma carnitine levels in comparison to those with normal cardiac function [98].

More recently, Li et al. [99] identified 39 biomarkers for detecting cardiotoxicity earlier than biochemical analysis and histopathological assessment. They used rats to create cardiotoxicity models in which the toxicity was caused by doxorubicin, isoproterenol, and 5-fluorouracil. The metabolomics analysis of plasma was performed using ultraperformance liquid chromatography quadrupole time-of-flight mass spectrometry. They used a support vector machine (SVM) to deploy a predictive model to confirm more exclusive biomarkers with more significant l-carnitine, 19-hydroxydeoxycorticosterone, lysophosphatidylcholine (LPC) (14:0), and LPC (20:2) [99].

Similarly, Schnackenberg et al. attempted to discover molecular markers of early-stage cardiotoxicity induced by doxorubicin in mice before the onset of cardiac damage. They discovered 18 metabolites significantly altered in the plasma, and another 22 metabolites were increased in cardiac tissue after a cumulative dose of 6 mg/kg, while myocardial injury and cardiac pathology were not noticed until after cumulative doses of 18 and 24 mg/kg, respectively [102]. Metabolomics analyses of plasma and heart tissue showed significant variations in the levels of many amino acids (including arginine and citrulline), biogenic amines, acylcarnitines (carnitine), and tricarboxylic acid cycle (TCA)-related metabolites (e.g., lactate, succinate).

Tan et al. conducted a study using gas chromatography–mass spectrometry to describe the metabolic profile of doxorubicin-induced cardiomyopathy in mice. They identified 24 metabolites, which were implicated in glycolysis, the citrate cycle, and the metabolism of some amino acids and lipids, and which were selected as possible biomarkers for the detection of cardiotoxicity [103].

Andreadou et al. used nuclear magnetic resonance (NMR) spectrometry to describe the metabolic profile of acute doxorubicin cardiotoxicity in rats and to evaluate the metabolic alterations conferred by co-treatment with oleuropein [90]. The mice were divided into six groups: the first group included the control group, the second group received DOX, and the other four groups of mice received doxorubicin with oleuropein in different doses and days, regarding the latter. Mice hearts were excised 72 h after doxorubicin administration and the H-NMR spectra of aqueous myocardium extracts were monitored. The results of the analysis showed an increase in the levels of acetate and succinate in the DOX group compared to the controls, while the amino acid levels were lower. The conclusion of the article was that acetate and succinate constituted novel biomarkers for the early detection of cardiotoxicity [100,105].

Geng et al. in their study, used gas chromatography−mass spectrometry analysis of the main targeted tissues (serum, heart, liver, brain, and kidney) to systemically evaluate the toxicity of DOX. Multivariate analyses revealed 21 metabolites in the serum, including cholesterol, d-glucose, d-lactic acid, glycine, l-alanine, l-glutamic acid, l-isoleucine, l-leucine, l-proline, l-serine, l-tryptophan, l-tyrosine, l-valine, N-methylphenylethanolamine, oleamide, palmitic acid, pyroglutamic acid, stearic acid, and urea, were changed in the serum in the DOX group [106].

Tantawy et al. identified a lower plasma abundance of pyruvate and a higher abundance of lactate in patients with carfilzomib-related cardiovascular adverse events. (CVAEs). They emphasized the significance of the pyruvate oxidation pathway associated with mitochondrial dysfunction. In order to better understand the mechanisms of carfilzomib-associated CVAEs, further investigation and validation are needed in a larger independent cohort [107].

Yin et al. proposed 15 different metabolites that play important roles in cyclophosphamide-induced cardiotoxicity. In their study, rat plasma samples were collected and analyzed one, three, and five days after cyclophosphamide administration using ultra-performance liquid chromatography quadrupole time-of-flight mass spectrometry (UPLC-QqTOF HRMS). Of the biomarkers studied, the proline, linoleic acid, and glycerophospholipids changed significantly in the three periods, and the changes were associated with an increasing time of occurrence of cardiotoxicity from cyclophosphamide [108].

The study of Jensen et al. [104] showed significant decreases in docosahexaenoic acid, arachidonic acid/eicosatetraenoic acid, o-phosphocolamine, and 6-hydroxynicotinic acid after sunitinib treatment with non-targeted metabolomics analysis of mice hearts [31]. The same author also showed alterations in the taurine/hypotaurine metabolism in the hearts and skeletal muscles of mice after sorafenib treatment [109].

Except for the analysis of plasma and heart tissue, NMR spectroscopy-based metabolomics may detect low-molecular-weight metabolites in urine and cell culture media. For example, Chaudhari et al. [101] showed a reduction in the utilization of pyruvate and acetate and an accumulation of formate in contrast to a control culture medium of human induced pluripotent stem cell-derived cardiomyocytes exposed to doxorubicin. In contrast, Wang et al. [59] showed in their study that tryptophan and phenylalanine metabolism in urine was also an important process in the systemic toxicity of doxorubicin. In addition, Park et al. identified 19 urinary metabolites in rats treated with doxorubicin [110].

This technology is still under development, but it seems obvious that metabolomics holds the potential to revolutionize our ability to profile samples in order to understand biological processes and find useful disease diagnostic biomarkers.

## 8. Conclusions

Cardiovascular toxicity continues to be a major cause of drug failure during preclinical and clinical treatment models and contributes to drug withdrawal after approval. Numerous medications that have been used frequently in adult clinical practice for a long time have demonstrated potentially harmful effects on the hearts of pediatric patients. The cardiotoxicity of these medications persists as a significant issue, having a negative impact on patients’ quality of life as well as their overall survival. Several strategies for the early detection of cardiotoxicity have been developed to reduce the number of patients with cardiac mortality and morbidity. Of importance, the biomarkers identified by the “omics” approach are considered new potential markers, especially in the scenario of diagnosis and the risk stratification of acute coronary syndromes induced by chemotherapeutic drugs, and they may prove helpful in the early detection of anticancer cardiotoxicity.

## Figures and Tables

**Figure 1 diagnostics-13-01864-f001:**
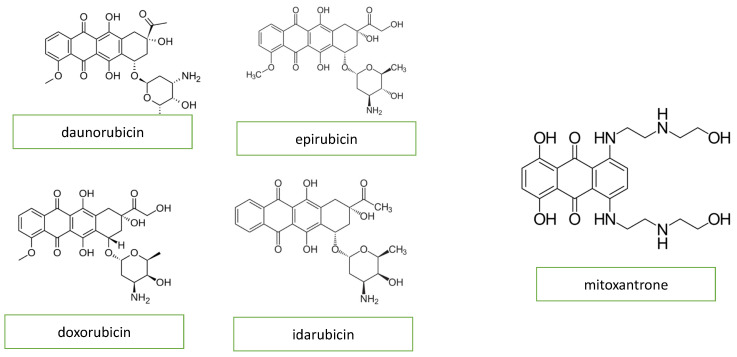
Chemical structure of anthracyclines and mitoxantrone.

**Figure 2 diagnostics-13-01864-f002:**
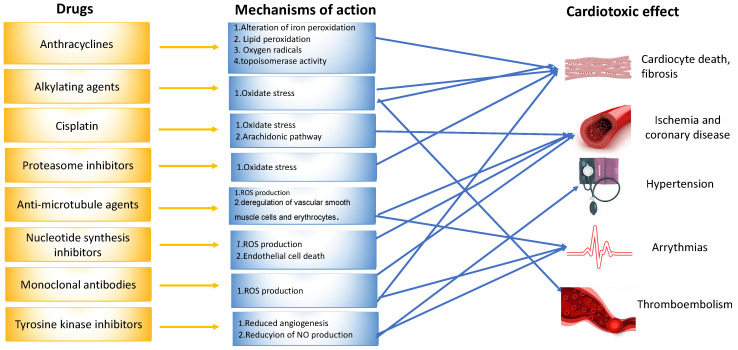
Mechanisms of drug-induced cardiotoxicity.

**Figure 3 diagnostics-13-01864-f003:**
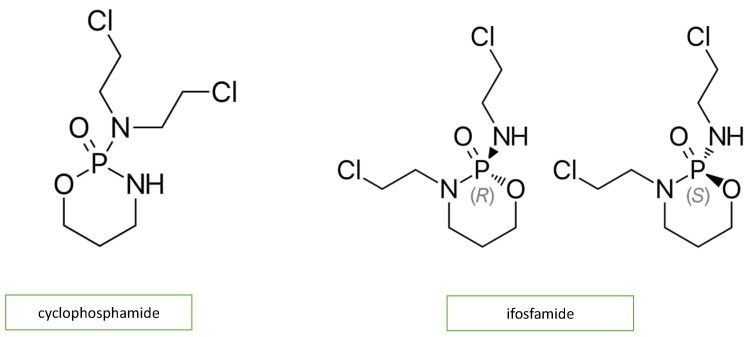
Chemical structure of alkylating agents.

**Figure 4 diagnostics-13-01864-f004:**
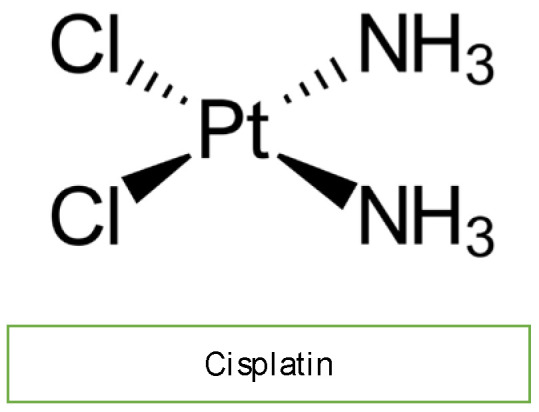
Chemical structure of cisplatin.

**Figure 5 diagnostics-13-01864-f005:**
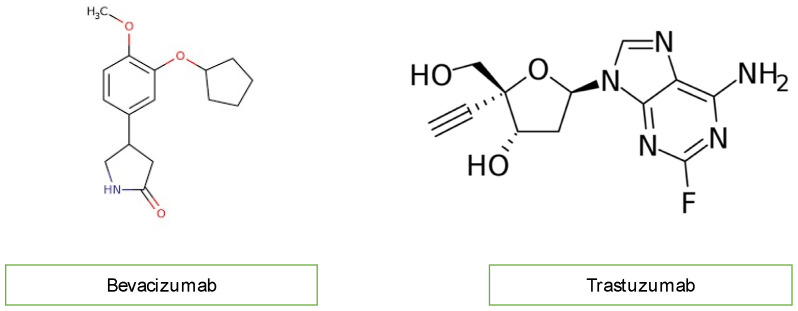
Chemical structure of monoclonal antibodies.

**Figure 6 diagnostics-13-01864-f006:**
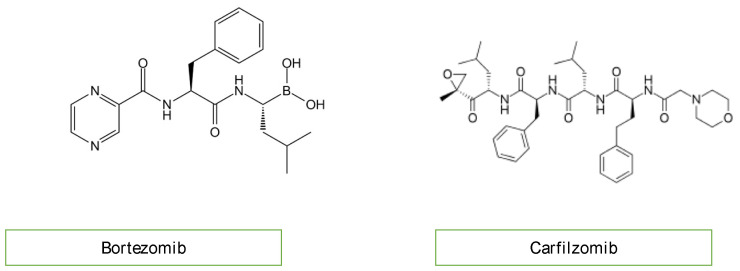
Chemical structure of proteasome inhibitors.

**Figure 7 diagnostics-13-01864-f007:**
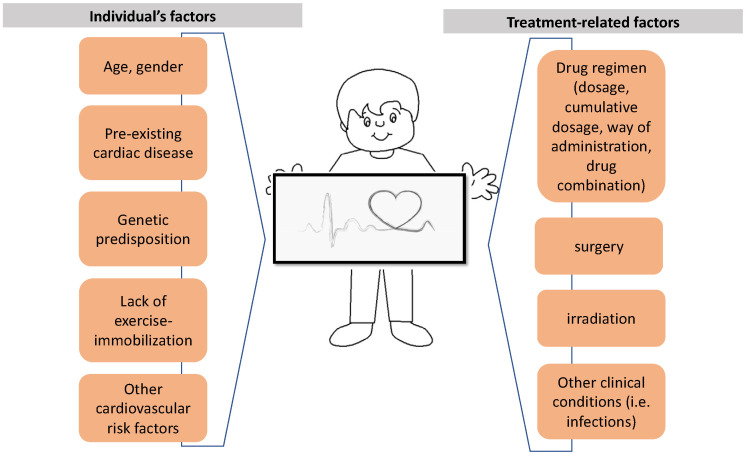
Multivariable factors leading to cardiotoxicity.

**Figure 8 diagnostics-13-01864-f008:**
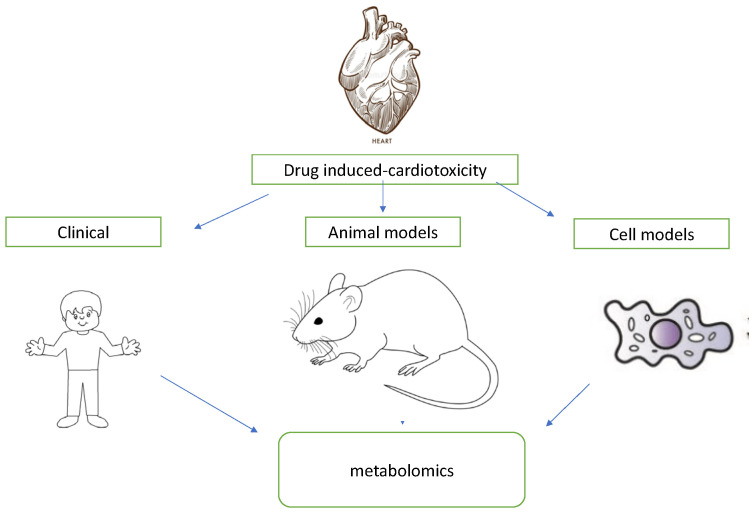
Diagram expressing the different research models of cardiotoxicity (clinical, animal, cellular) using metabolomic data.

**Table 1 diagnostics-13-01864-t001:** Chemotherapeutic drugs and cardiovascular toxicity. (Adjusted by Rochette et al., Trends Pharmacol Sci. 2015 Jun;36(6):326–48 [25]).

Medicine/Cardiotoxicity	Incidence (%)	Arrhythmia	Myocardial Ischemia	Vascular Toxicity	Heart Failure	QT Prolongation	Arterial Hypertension
**Anthracyclines**							
Doxorubicin	3–26	xxx	x	NE	xxx	NE	x
Doxorubicin Liposomal	2	x	xx	NE	x	NE	x
Epirubicin	0.9–3.3	x	x	NE	x	NE	x
Daunorubicin		xx	x	NE	x	NE	x
Idarubicin	5–18	xxx	x	NE	xx	NE	x
**Antibiotics**							
Mitoxantrone	0.2–30	xxx	xx	NE	xx	NE	xx
Mitomycin-c	10	xx	xx	NE	xx		NE
**Monoclonal antibodies**							
Trastuzumab	1.7–8	xx	x	xx	xxx	NE	xx
Bevacizumab	1.6–4	xx	xx	xxx	xx	NE	xx
Pertuzumab	0.7–1.2	x	x	x	xx	NE	x
Dinutuximab beta		NE	xx	NE	xx	NE	xx
Rituximab		x	xx	xxx	x	NE	xx
**Tyrosine kinase inhibitors**							
Dasatinib	2–4	xxx	xx	xx	xx	xx	xx
Nilotinib	1	xx	NE	x	xx	xx	xxx
Vermurafenib		xx	xx	xx	x	NE	xx
Sorafenib	2–28	x	xx	xx	xx	NE	xx
Sunitinib	2.7–15	x	xx	xx	xxx	x	xxx
Erlotinib	7–11	NE	xx	xx	NE	NE	NE
Lapatinib	0.2–1.5	NE	xx	x	NE	xxx	NE
Pazopanib	7–11	NE	xx	xx	x	NE	xxx
Imatinib	0.2–2.7	NE	xxx	xx	xx	NE	NE
**Proteasome inhibitors**							
Bortezomib	2–5	x	x	x	x	NE	x
Carfilzomib	11–25	xx	xx	NE	x	NE	x
**Antimetabolites**							
5-fluorouracil	2–20	xxx	xxx	NE	x	NE	NE
Capecitabine		xxx	xxx	xx	NE	NE	NE
Clofarabine	27					NE	
**Alkylating agents**							
Cyclophosphamide	7–28	NE	NE	x	NE	NE	NE
Ifosfamide	0.5–17	NE	NE	x	xx	NE	NE
**Cisplatin**	rare	NE	NE	xx	NE	NE	NE
**Antimicrotubule agents**							
Paclitaxel	<1	xx	x	NE	x	NE	x
Docetaxel	2.3–13	xx	xx	NE	x	NE	xx
**Alkaloids of vinca**							
Vincristine	25	xx	x	NE	NE	xx	x
Vinblastine		NE	x	NE	NE	NE	x
Vindesin		NE	NE	NE	NE	NE	NE
Vinorelbin		NE	x	NE	NE	NE	NE

xxx: means >10%, xx: means 1–10%, x: means <1%, NE: not established.

**Table 2 diagnostics-13-01864-t002:** Risk factors.

Risk Factors Related to Children	Risk Factors Related to Therapy
Female sex Age < 5 years Genetic background Pre-existing cardiac disease Cardiovascular risk factors (diabetes, obesity, hyperlipidemia, hypertension)	Anthracycline > 250 mg/m^2^ equivalent doxorubicin Cumulative dose Irradiation Combination of cardiotoxic drugs

**Table 3 diagnostics-13-01864-t003:** Summary of major miRNAs link to drug-induced cardiotoxicity in people.

MiRNA	Drug	Modulation	Species	System	References
miR-1	Doxorubicin	Increase	Female patients	Plasma	Riguad et al., Oncotarget 2017 [84]
miR-1, miR-29b, miR-499	Anthracyclines	Increase	Children and young adult	Plasma	Leger et al., J Am Heart Assoc. 2017 [85]
miR1254	Bevacizumub	Increase	Humans	Plasma	Zhao et al., Tumour Biol. 2014 [89]
miR29 miR499	Doxorubicin	Increase	Children	Plasma	Oatmen et al., Am J Physiol Heart Circ Physiol, 2018 [73]
miR208	Doxorubicin	Nothing	Female patients	Plasma	Carvalho et al., J Appl Toxicol 2015 [74]

**Table 4 diagnostics-13-01864-t004:** Metabolomics associated with drug-induced cardiotoxicity.

Metabolite	Plasma	Stem Cell	Heart	Mice	People	XRT	Medicine	Dose	Biomarker	References
Proline	↓//↑		↑	Yes	No		Cyclophosphamide	200 mg/kg		Li et al., J Proteome Res, 2015 [99]
LPC 20:3	↓			Yes	No		Cyclophosphamide	200 mg/kg		Li et al., J Proteome Res, 2015 [99]
Linoleic acid	↓			Yes	No		Cyclophosphamide	200 mg/kg		Li et al., J Proteome Res, 2015 [99]
L-carnitine	↑//↑			Yes	No		Cyclophosphamide/doxo/isoprotenerol/5-fluorouracil	200 mg/kg//20 mg/kg//5 mg/kg//125 mg/kg		Li et al., J Proteome Res, 2015 [99]
19-hydroxycorticosterone	↑//↓			Yes	No		Cyclophosphamide/doxo/isoprotenerol/5-fluorouracil	200 mg/kg//20 mg/kg//5 mg/kg//125 mg/kg		Li et al., J Proteome Res, 2015 [99]
Phytophingosine	↓			Yes	No		Cyclophosphamide	200 mg/kg		Li et al., J Proteome Res, 2015 [99]
Cholid acid	↓			Yes	No		Cyclophosphamide	200 mg/kg		Li et al., J Proteome Res, 2015 [99]
LPC 14:0	↓//↓			Yes	No		Cyclophosphamide/doxo/isoprotenerol/5-fluorouracil	200 mg/kg//20 mg/kg//5 mg/kg//125 mg/kg		Li et al., J Proteome Res, 2015 [99]
LPC 18:3	↓			Yes	No		Cyclophosphamide	200 mg/kg		Li et al., J Proteome Res, 2015 [99]
LPC 16:1	↓			Yes	No		Cyclophosphamide	200 mg/kg		Li et al., J Proteome Res, 2015 [99]
LPE 18:2	↓			Yes	No		Cyclophosphamide	200 mg/kg		Li et al., J Proteome Res, 2015 [99]
LPC 22:5	↓			Yes	No		Cyclophosphamide	200 mg/kg		Li et al., J Proteome Res, 2015 [99]
LPC 22:6	↓			Yes	No		Cyclophosphamide	200 mg/kg		Li et al., J Proteome Res, 2015 [99]
LPC 22:4	↓			Yes	No		Cyclophosphamide	200 mg/kg		Li et al., J Proteome Res, 2015 [99]
LPC 20:2	↓//↓			Yes	No		Cyclophosphamide/doxo/isoprotenerol/5-fluorouracil	200 mg/kg//20 mg/kg//5 mg/kg//125 mg/kg		Li et al., J Proteome Res, 2015 [99]
PLE 20:3	↓			Yes	No		Cyclophosphamide	200 mg/kg		Li et al., J Proteome Res, 2015 [99]
Pyruvate		↑					Doxorubicin	20 mg/kg	Troponin T LDH	Andreadou et al., ΝΜR Biomed, 2009 [100] /Chaudhari et al., Amino Acids 2017 [101]
Acetate		↑	↑	Yes			Doxorubicin	20 mg/kg	Troponin T LDH	Andreadou et al., ΝΜR Biomed, 2009 [100] /Chaudhari et al., Amino Acids 2017 [101]
Formate		↑					Doxorubicin	20 mg/kg	Troponin T LDH	Andreadou et al., ΝΜR Biomed, 2009 [100] /Chaudhari et al., Amino Acids 2017 [101]
Succinate		↑	↑	Yes			Doxorubicin	20 mg/kg	Troponin T LDH	Andreadou et al., ΝΜR Biomed, 2009 [100] /Chaudhari et al., Amino Acids 2017 [101]
Lactate	↑//↑		↓	Yes			Doxorubicin	20 mg/kg	Troponin T	Andreadou et al., ΝΜR Biomed, 2009 [100]
Alanine	↑//↑		↑//↑	Yes			Doxorubicin	20 mg/kg	Troponin T	Andreadou et al., ΝΜR Biomed, 2009 [100]
Glutamine	↑		↓	Yes			Doxorubicin	20 mg/kg	Troponin T	Andreadou et al., ΝΜR Biomed, 2009 [100]
Glutamate	↑		no	Yes		↑	Doxorubicin	20 mg/kg	Troponin T	Andreadou et al., ΝΜR Biomed, 2009 [100]
Creatine			no	Yes			Doxorubicin	20 mg/kg	Troponin T	Andreadou et al., ΝΜR Biomed, 2009 [100]
Taurine			no	Yes		↓	Doxorubicin	20 mg/kg	Troponin T	Andreadou et al., ΝΜR Biomed, 2009 [100]
Valine	↑		↓	Yes		↑	Doxorubicin	20 mg/kg	Troponin T	Andreadou et al., ΝΜR Biomed, 2009 [100]
Leuline	↑		↓	Yes			Doxorubicin	20 mg/kg	Troponin T	Andreadou et al., ΝΜR Biomed, 2009 [100]
Isoleukine	↑		↓	Yes		↑	Doxorubicin	20 mg/kg	Troponin T	Andreadou et al., ΝΜR Biomed, 2009 [100]
Carnitine	↓//↑		↓	Yes	**Yes**		Anthracyclines/doxorubicin		Troponin T	Armenian et al., Cancer Epidemiol Biomarkers Prev. 2014 [98]
Threitol	↓				Yes		Anthracyclines			Armenian et al., Cancer Epidemiol Biomarkers Prev. 2014 [98]
Mannose	↓				Yes		Anthracyclines			Armenian et al., Cancer Epidemiol Biomarkers Prev. 2014 [98]
Pyroglutamine	↓				Yes		Anthracyclines			Armenian et al., Cancer Epidemiol Biomarkers Prev. 2014 [98]
N-acetylalanine	↓				Yes		Anthracyclines			Armenian et al., Cancer Epidemiol Biomarkers Prev. 2014 [98]
Creatine	↓				Yes		Anthracyclines			Armenian et al., Cancer Epidemiol Biomarkers Prev. 2014 [98]
Eicosenoate	↓				Yes		Anthracyclines			Armenian et al., Cancer Epidemiol Biomarkers Prev. 2014 [98]
Stearidonate	↓				Yes		Anthracyclines			Armenian et al., Cancer Epidemiol Biomarkers Prev. 2014 [98]
Arachidonate	↓				Yes		Anthracyclines			Armenian et al., Cancer Epidemiol Biomarkers Prev. 2014 [98]
Dihomo-linoleate	↓				Yes		Anthracyclines			Armenian et al., Cancer Epidemiol Biomarkers Prev. 2014 [98]
L-stearoylglcerophoinositol	↓				Yes		Anthracyclines			Armenian et al., Cancer Epidemiol Biomarkers Prev. 2014 [98]
Dehydroisoandrosterone sulfate	↓				Yes		Anthracyclines			Armenian et al., Cancer Epidemiol Biomarkers Prev. 2014 [98]
Pregnen-dio; disulfate	↓				Yes		Anthracyclines			Armenian et al., Cancer Epidemiol Biomarkers Prev. 2014 [98]
Pregn steroid monosulfate	↓				Yes		Anthracyclines			Armenian et al., Cancer Epidemiol Biomarkers Prev. 2014 [98]
Arginine	↑		↑	Yes			Doxorubicin			Schnackenberg et al., Appl. Toxicol. 2016 [102]
Asparagine	↑		↑	Yes			Doxorubicin		Troponin T	Schnackenberg et al., Appl. Toxicol. 2016 [102]
Citrulline	↑		↑	Yes			Doxorubicin		Troponin T	Schnackenberg et al., Appl. Toxicol. 2016 [102]
Glycine	↑		↑	Yes		↑	Doxorubicin		Troponin T	Schnackenberg et al., Appl. Toxicol. 2016 [102]
Histidine	↑		↑	Yes			Doxorubicin		Troponin T	Schnackenberg et al., Appl. Toxicol. 2016 [102]
Lysine	↑		↑	Yes			Doxorubicin		Troponin T	Schnackenberg et al., Appl. Toxicol. 2016 [102]
Methionine	↑		↑	Yes			Doxorubicin		Troponin T	Schnackenberg et al., Appl. Toxicol. 2016 [102]
Ornithine	↑		↑	Yes			Doxorubicin		Troponin T	Schnackenberg et al., Appl. Toxicol. 2016 [102]
Phenylalanine	↑		↑	Yes			Doxorubicin		Troponin T	Schnackenberg et al., Appl. Toxicol. 2016 [102]
Serine	↑		↑	Yes			Doxorubicin		Troponin T	Schnackenberg et al., Appl. Toxicol. 2016 [102]
Threonine	↑		↑	Yes		↑	Doxorubicin		Troponin T	Schnackenberg et al., Appl. Toxicol. 2016 [102]
Tryptophan	↑		↑	Yes			Doxorubicin		Troponin T	Schnackenberg et al., Appl. Toxicol. 2016 [102]
Tyrosine	↑		↑	Yes			Doxorubicin		Troponin T	Schnackenberg et al., Appl. Toxicol. 2016 [102]
Acetylornithine	↑		↓	Yes			Doxorubicin		Troponin T	Schnackenberg et al., Appl. Toxicol. 2016 [102]
Hydroxproline	↑		No	Yes			Doxorubicin		Troponin T	Schnackenberg et al., Appl. Toxicol. 2016 [102]
Citrate	no		No	Yes			Doxorubicin		Troponin T	Schnackenberg et al., Appl. Toxicol. 2016 [102]
Propionylcarnitine	↑		No	Yes			Doxorubicin		Troponin T	Schnackenberg et al., Appl. Toxicol. 2016 [102]
Serotonine	no		↑	Yes			Doxorubicin		Troponin T	Schnackenberg et al., Appl. Toxicol. 2016 [102]
Putrescine	no		↑	Yes			Doxorubicin		Troponin T	Schnackenberg et al., Appl. Toxicol. 2016 [102]
Malate	↑		↑	Yes			Doxorubicin			Tan et al., PLoS ONE 2011 [103]
Fructose			↑	Yes			Doxorubicin			Tan et al., PLoS ONE 2011 [103]
Glycose			↑	Yes			Doxorubicin			Tan et al., PLoS ONE 2011 [103]
Cholesterol			↑	Yes			Doxorubicin			Tan et al., PLoS ONE 2011 [103]
Alanine			↑	Yes			Doxorubicin			Tan et al., PLoS ONE 2011 [103]
Glutamine				Yes		↓	Doxorubicin			Tan et al., PLoS ONE 2011 [103]
Docosahexaenoic acid			↓	Yes			Sunitinib			Jensen et al., Metabolites. 2017 [104]
Arachidonic acid/eicosapetaenoic acid			↓	Yes			Sunitinib			Jensen et al., Metabolites. 2017 [104]
6-hydroxynicotinic acid			↓	Yes			Sunitinib			Jensen et al., Metabolites. 2017 [104]
O-phosphocolamine			↓	Yes			Sunitinib			Jensen et al., Metabolites. 2017 [104]
Ethanolamine	↑			Yes			Sunitinib			Jensen et al., Metabolites. 2017 [104]
Xenobiotics										

## Data Availability

The article is a review, and no original data were generated.

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
