# Peer review of "Prognostic Factors for Cardiotoxicity among Children with Cancer: Definition, Causes, and Diagnosis with Omics Technologies"

_diagnostics, 2023, doi:10.3390/diagnostics13111864_

Round 1

Reviewer 1 Report

Prognostic factors for cardiotoxicity among children with cancer: definition, causes and diagnosis with omics’ technologies’'

The authors' proposal is interesting, however several changes must be made to improve the writing and thus the understanding of the work.

1.       Abstract: Authors must rewrite the abstract and organize the information in such a way as to make each part of an abstract clear. Introduction, hypothesis, and the objective of the work, not to clear. So deeply focus the factors on cardiotoxicity.

2.       Introduction:  authors described more detailed about  anthracyclines and mitoxantrone, alkylating agents, proteasome inhibitors etc.,. Authors should describe more detail in diagnosis.

3.       Should include small paragraph role of cisplatin on cardiac hypertrophy.

4. If possible put images 

Moderate editing of English language required

Author Response

Dear reviewer,

Thank you for your revisions in order to improve my manuscript.

As per your suggestions, the following changes have been made:

  1.       Abstract: Authors must rewrite the abstract and organize the information in such a way as to make each part of an abstract clear. Introduction, hypothesis, and the objective of the work, not to clear. So deeply focus the factors on cardiotoxicity.------> Abstract has been remodeled as indicated.

    Improvements in the treatment of childhood cancer have considerably enhanced survival rates over the last decades, to over 80% as per today. However, this great achievement has been accompanied by the occurrence of several early and long-term treatment-related complications major of which is the cardiotoxicity.

    This article reviews the contemporary definition of cardiotoxicity, older and newer chemotherapeutic agents that are mainly involved in cardiotoxicity, routine process diagnosis and the methods using omics technology for early and preventive diagnosis.

    The chemotherapeutic agents and radiation therapies have been implicated as a cause of cardiotoxicity. In response, the area of cardio-oncology has developed into a crucial element of oncologic patient care, committed to the early diagnosis and treatment of adverse cardiac events. However, routine diagnosis and monitoring of cardiotoxicity rely on electrocardiography and echocardiography. For the early detection of cardiotoxicity, in recent years, major studies have been conducted using biomarkers such as troponin, N-terminal pro b-natriuretic peptide, etc. Despite refinements in diagnostics still severe limitations exist, due to the increase of the above-mentioned biomarkers, only after significant cardiac damage has occurred. Lately, research has been expanded by introducing new technologies and finding new markers by omics approach. These new markers could be used not only for early detection, but also for early prevention of cardiotoxicity.

    The omics science, which includes genomics, transcriptomics, proteomics, and metabolomics, offers new opportunities for biomarker discovery in cardiotoxicity and may provide understanding of mechanisms of cardiotoxicity, beyond traditional technologies.

  2. Introduction:  authors described more detailed about  anthracyclines and mitoxantrone, alkylating agents, proteasome inhibitors etc.,. Authors should describe more detail in diagnosis.------>we added a paragraph describe with more detail the methods of diagnosis of cardiotoxicity:

    As part of the baseline risk assessment, thorough clinical history and physical examination are advised. Many patients with cardiac dysfunction could be asymptomatic, so both during and post chemotherapy, cardiac monitoring is necessary. Consideration of the classic cardiovascular disease risk factors already mentioned is recommended and children should be monitored for clinical signs and potential indicators for cardiotoxicity. The above mentioned factors should be noted alongside baseline electrocardiography, cardiac serum biomarkers, and cardiac imaging tests to complete baseline evaluation.

  3. Should include small paragraph role of cisplatin on cardiac hypertrophy----> we added a small paragraph regarding cisplatin Cisplatin cardiotoxicity can be caused by reactive oxygen species generation, which leads to the creation of oxidative stress and endothelial capillary damage (vascular damage), or by a direct toxic effect on cardiac myocytes. [10] The prolonged cardiovascular toxicity of cisplatin, lasting up to many years, has been explained by both direct diffuse endothelial damage and an increase in risk factors for cardiovascular disease. These effects include coronary artery disease, systolic or diastolic left ventricular dysfunction, and severe congestive cardiomyopathy.
  4. If possible put images -----? images have been added

Reviewer 2 Report

diagnostics-2349310

The manuscript: childhood cancer have considerably enhanced survival rates over the last decades, to over 80%. The authors take a look chemotherapeutic agents that have been implicated as a cause of cardiotoxicity( anthracyclines and mitoxantrone, alkylating agents, proteasome inhibitors; anti-microtubule agents, cisplatin, monoclonal antibodies, small molecular tyrosine kinase inhibitors, and nucleotide synthesis inhibitors) and the studies using biomarkers such as troponin, N-terminal pro b-natriuretic peptide for investigation for this damage. The article explane the using omics technology, as a new markers could be used not only for early detection, but also for early prevention of cardiotoxicity, and as methods of early and preventive diagnosis. 

I have no remarks on the individual parts of the manuscript. I have a few suggestions that I think will improve readability and visualization:

1. To add the chemical structure of the used agents;

2.Schematic diagram of the mechanisms of drugs-induced cardiotoxicity with different sections: the suppressesion the function; ROS production induces;  pro-apoptotic proteins; activities of Ca2+ handling proteins; iron overload, apoptosis, and ferroptosis, fibrosis; receptor activation.

3. Schematic diagram of the expression the different researche models of cardiotooxicity- clinical, animal, cellular;

4.  Renewal some of the references from the last 5 years- only 4 % of the references are from the last 5 years.  

Minor editing of English language required.

Author Response

Dear reviewer,

thank you for your suggestions in order to improve my manuscript.

1. To add the chemical structure of the used agents;-----> chemical structures of the following agents were added: anthracyclines, mitoxantrone, proteasome inhibitors, cisplatin, alkylating agents and monoclonal antibodies.

2.Schematic diagram of the mechanisms of drugs-induced cardiotoxicity with different sections: the suppressesion the function; ROS production induces;  pro-apoptotic proteins; activities of Ca2+ handling proteins; iron overload, apoptosis, and ferroptosis, fibrosis; receptor activation.--->mechanism diagrams were added

3. Schematic diagram of the expression the different research models of cardiotooxicity- clinical, animal, cellular;---> relative diagram was added

4.  Renewal some of the references from the last 5 years- only 4 % of the references are from the last 5 years.----->the following references were renewed: 10,11,36,41

Round 2

Reviewer 1 Report

Satisfy with author comments

Moderate editing of English language